# Facile and Controllable Synthesis of Large-Area Monolayer WS_2_ Flakes Based on WO_3_ Precursor Drop-Casted Substrates by Chemical Vapor Deposition

**DOI:** 10.3390/nano9040578

**Published:** 2019-04-09

**Authors:** Biao Shi, Daming Zhou, Shaoxi Fang, Khouloud Djebbi, Shuanglong Feng, Hongquan Zhao, Chaker Tlili, Deqiang Wang

**Affiliations:** 1Chongqing Key Lab of Multi-Scale Manufacturing Technology, Chongqing Institute of Green and Intelligent Technology, Chinese Academy of Sciences, Chongqing 400714, China; shibiao@cigit.ac.cn (B.S.); dmzhou@cigit.ac.cn (D.Z.); fangshaoxi@cigit.ac.cn (S.F.); khouloud@cigit.ac.cn (K.D.); fengshuanglong@cigit.ac.cn (S.F.); hqzhao@cigit.ac.cn (H.Z.); 2University of Chinese Academy of Sciences, Beijing 100049, China

**Keywords:** WS_2_, 2D materials, large-area, CVD, fluorescence emission, Raman mapping

## Abstract

Monolayer WS_2_ (Tungsten Disulfide) with a direct-energy gap and excellent photoluminescence quantum yield at room temperature shows potential applications in optoelectronics. However, controllable synthesis of large-area monolayer WS_2_ is still challenging because of the difficulty in controlling the interrelated growth parameters. Herein, we report a facile and controllable method for synthesis of large-area monolayer WS_2_ flakes by direct sulfurization of powdered WO_3_ (Tungsten Trioxide) drop-casted on SiO_2_/Si substrates in a one-end sealed quartz tube. The samples were thoroughly characterized by an optical microscope, atomic force microscope, transmission electron microscope, fluorescence microscope, photoluminescence spectrometer, and Raman spectrometer. The obtained results indicate that large triangular monolayer WS_2_ flakes with an edge length up to 250 to 370 μm and homogeneous crystallinity were readily synthesized within 5 min of growth. We demonstrate that the as-grown monolayer WS_2_ flakes show distinctly size-dependent fluorescence emission, which is mainly attributed to the heterogeneous release of intrinsic tensile strain after growth.

## 1. Introduction

The isolation and synthesis of atomically thin two-dimensional transition metal dichalcogenides (TMDs), such as MS_2_ (M = Mo, W), have attracted a lot of interest due to their unprecedented properties compared with their bulk counterparts [1,2,3]. Unlike graphene and h-BN (h-Born Nitride), atomically thin MS_2_ possess semiconducting behavior with an intrinsic direct-energy gap corresponding to the visible frequency range, and show strong spin-orbit coupling and band splitting [4]. Of these, molybdenum disulfide (MoS_2_) has received tremendous attention because of its unique crystal structure, and physical, chemical, and electrical properties [5,6,7,8,9,10,11,12]. Due to these fascinating properties, MoS_2_ has been used in numerous applications, including photodetection, gas-sensing, and hydrogen evolution [13,14,15,16,17]. Atomically thin tungsten disulfide (WS_2_) has a similar structure to MoS_2_, but exhibits stronger photoluminescence (PL) quantum yield at room temperature and larger spin-orbit coupling compared to MoS_2_ [18,19,20], which makes it promising for applications in new optoelectronics and spintronics. Nevertheless, the research on WS_2_ lags far behind that of MoS_2_ and the development of simple and controllable strategies for preparing high-quality and large-area atomically thin WS_2_ is still a big challenge, which has impeded further fabrication of optoelectronics/spintronics devices.

As reported, large-area continuous mono- and multi-layer WS_2_ were synthesized by sulfurizing the pre-deposited ultrathin tungsten or WO_x_ films obtained by various routes, including thermal evaporation, magnetron sputtering, and atomic layer deposition (ALD) [21,22,23,24,25,26,27]. This two-step chemical vapor deposition (CVD) method can provide large-area polycrystalline WS_2_ film for the fabrication of devices. However, the size of the single-crystal domain in the polycrystalline film is limited to the nanoscale and its electrical properties could be seriously degraded because of the existence of abundant grain boundaries (GBs) [28]. Moreover, the pre-deposition of ultrathin films is time-consuming and more efforts should be paid to precisely control the thickness, which restricts the wide use of two-step CVD in research communities. To simplify the synthesis process, one-step CVD that is based on direct sulfurization of powdered WO_3_ on diverse substrates, such as SiO_2_/Si, sapphire, Au foils, and *h*-BN, has been employed for the synthesis of WS_2_ [29,30,31,32,33,34]. In this direction, powdered sulfur and WO_3_ in quartz/ceramic boats are placed at separate locations; normally the sulfur is located upstream and the WO_3_ is placed underneath or ahead of the substrates. In addition, growth parameters, such as the temperature, heating rate, gas flow, and the distance between the precursor and substrate, can be accurately controlled between batches. However, this strategy presents some drawbacks since the distribution of the solid-phase precursors loaded in quartz/ceramic boats is not easy to control, and this could significantly influence the concentration of gaseous species on the growth interface. Furthermore, due to the difference in melting points of sulfur and WO_3_, it is difficult to simultaneously and precisely control the evaporation of these two precursors and the subsequent transportation to the growth interface. As a result, the dimension and morphology of the CVD-grown WS_2_ always present significant differences even under the same growth parameters. 

Lee et al. [35] reported that continuous polycrystalline monolayer WS_2_ films in the centimeter-scale were grown on SiO_2_/Si substrate with the seeding of perylene-3,4,9,10-tetracarboxylic acid tetrapotassium salt (PTAS). However, the size of the isolated single-crystal domains was limited in the range of 10 to 20 μm. Zhang et al. [31] reported the synthesis of mono- and multi-layer WS_2_ flakes with the domain size exceeding 50 × 50 μm^2^ on the sapphire substrate at low-pressure with mixed hydrogen and argon gases. Yue et al. [36] reported that uniform triangular monolayer WS_2_ flakes with a side length of ~233 μm were synthesized on SiO_2_/Si substrate by carefully adjusting the introduction time of the sulfur precursor and the distance between the sources and substrates. Recently, Zhou et al. [37] demonstrated that molten-salt-assisted CVD can be used to synthesize large-area triangular monolayer WS_2_ flakes with an edge length of 300 μm at moderate temperatures due to the reduction of the WO_3_ melting point by using sodium chloride. To this end, significant attempts have been made towards the large-area synthesis of monolayer WS_2_ by one-step CVD. However, the layer controllability and universality of the synthesis method are the major existing obstacles that prevent practical applications. Therefore, controllable synthesis of large-area monolayer WS_2_ is still challenging. 

In this study, we report a facile and controllable method for synthesis of large-area monolayer WS_2_ flakes by one-step CVD at atmospheric pressure. To promote the distribution of WO_3_, we first dispersed the powdered WO_3_ in ethanol to form a suspension solution, then we used a pipette to drop-cast the solution onto the SiO_2_/Si substrates. Moreover, we simultaneously loaded the powdered precursors and SiO_2_/Si substrates in a small one-end sealed quartz tube; and we placed the small tube inside a bigger quartz tube to ensure adequate amounts of sulfur-precursors to participate in the whole process of WS_2_ growth. Through this method, we obtained a series of triangular monolayer WS_2_ flakes with edge lengths up to 250 to 370 μm and homogeneous crystallinity. The morphology, thickness, atomic structure, and light emissions of the as-grown WS_2_ samples were characterized by various tools, such as an optical microscope (OM), atomic force microscope (AFM), transmission electron microscope (TEM), fluorescence (FL) microscope, PL spectrometer, and Raman spectrometer. It was found that the as-grown monolayer WS_2_ flakes with edge lengths greater than 95 μm show suppressed fluorescence emission in the inner region, while the smaller one presents homogeneous fluorescence emission. Raman mapping results indicate that the distinctly size-dependent fluorescence emission is attributed to the heterogeneous release of intrinsic tensile strain within WS_2_ after growth. Furthermore, the results of five independent experiments are consistent with each other, confirming the validity and reproducibility of this method.

## 2. Materials and Methods

In this research, the synthesis of WS_2_ was achieved by one-step CVD in a horizontal single-zone furnace (Hefei FACEROM Co., Ltd., Hefei, China) at atmospheric pressure, as shown in Figure 1a. The CVD system mainly consists of the heating zone and a quartz tube with a 60 mm diameter. Upstream of the quartz tube is connected to the high-purity (99.999%) argon cylinder, while downstream is connected to the exhausted gas treatment system. We did not use hydrogen in this research, due to considerations of safety, although previous reports demonstrated that hydrogen could facilitate the reduction of WO_3_ and lead to the growth of high quality WS_2_ [38,39]. Furthermore, a small quartz tube with a diameter of 15 mm sealed at one-end was intentionally employed for holding precursors and substrates simultaneously, this procedure was different from previous reports [36,40,41], aiming at obtaining large-area WS_2_ in a short period of time.

The SiO_2_ (300 nm)/Si (5 mm × 5 mm) sliced from a 4-inch wafer was used as substrate and was sequentially washed in DI (Deionized) water, acetone, DI water, ethanol, and DI water in an ultrasonic bath for 10 min each. The residual water and organic solvent were removed by compressed UHP (Ultra-high Purity) nitrogen gas blowing. Alcoholic tungsten suspension solution with a concentration of 1.5 mg/mL was prepared by WO_3_ (>99.9%, Adamas-beta, Shanghai, China) and ethanol (AR, KESHI, Chengdu, China). Before the growth process, a drop (10 μL) of alcoholic tungsten suspension solution was directly dropped onto the cleaned substrate using a pipette, as shown in Figure 1b, and the substrate was dried on a heater (40 °C) for 3 min. A relatively uniform distribution of powdered WO_3_ on the SiO_2_/Si substrate was obtained (as shown in Appendix A) after the evaporation of ethanol. Then, another fresh and cleaned SiO_2_/Si substrate was placed face-down above the drop-casted substrate to form a sandwich-structure. Subsequently, 600 mg of sulfur powder (S sublimed, >99.5%, KESHI, Chengdu, China) was introduced at the bottom of the sealed end of the small quartz tube, while the sandwich-structure substrates were carefully placed downstream, 24 cm away from the powdered sulfur. After that, the small quartz tube together with the precursors and substrates was loaded into the bigger quartz tube carefully, ensuring the substrates were exactly located at the center of the heating zone. Before heating, the furnace chamber was pumped to low pressure (<10 Pa) and then purged by 200 sccm argon to atmospheric pressure. Figure 1c shows the temperature profile used in this study. Initially, the furnace was heated from room temperature to 200 °C with a heating rate of 10 °C/min to remove the contaminants, such as water or residual organics. Subsequently, a higher heating rate of 28 °C/min was used to increase the temperature to 950 °C to obtain a high nucleation rate of WS_2_. The growth of WS_2_ was kept at 950 °C for 5 min under 200 sccm of argon. After growth, the furnace was cooled naturally to room temperature. 

The morphology and size of the as-grown WS_2_ samples were characterized by an OM (50i POL, Nikon, Tokyo, Japan). The thickness and atomic structure of the as-grown WS_2_ were analyzed via an AFM (Dimension EDGE, Bruker, Billerica, MA, USA) in tapping mode and TEM (Tecnai G^2^ F20, FEI, Hillsboro, OR, USA), respectively. Raman spectra were collected in the backscattering geometry at room temperature with an excitation wavelength of 532 nm (InVia Reflex, Renishaw, Gloucestershire, UK). Before Raman characterization, the system was calibrated with the Raman peak of Si at 520 cm^−1^. Photoluminescence spectra were obtained from a home-made laser scanning confocal microscope system. A 532 nm CW laser with an average power of 100 μW was employed as the excitation source to avoid sample damage. Fluorescence images were captured by using a fluorescence microscope (BX53, Olympus, Tokyo, Japan) under excitation of a green light source.

## 3. Results and Discussion

Figure 2a shows a typical optical image of the as-grown WS_2_ sample on the covering SiO_2_/Si substrate, which is blank without WO_3_-ethanol drop-casting. The as-grown WS_2_ exhibits the feature of an equilateral triangle with sharp edges. Furthermore, the GBs can also be observed, which is caused by the coalescence of adjacent triangles and is hard to avoid in CVD-grown MoS_2_ and WS_2_ flakes [6,28]. It is noted that all the triangles regardless of their sizes show uniform contrast under OM, indicating good uniformity of the thickness of the as-grown WS_2_. Figure 2b presents a representative optical image of the WS_2_ sample grown on the bottom SiO_2_/Si substrate, which was drop-casted by WO_3_-ethanol solution. In contrast with the covering substrate, small triangular WS_2_ with uniform contrast can be occasionally observed, however, irregular-shape WS_2_ with heterogeneous contrast and particulate WS_2_ (as shown in Appendix A) are predominant on the bottom substrate. These distinctly morphological features could be caused by the existence of the powdered WO_3_ and highly concentrated reaction sources on the bottom substrate [33]. Due to the differences in the nucleation density and local environment on the substrate, the dimension distribution of the triangular flakes is quite wide, from tens to hundreds of micrometers, while large triangular WS_2_ flakes with edge lengths of ~270 μm are readily obtained on the covering SiO_2_/Si substrate, as shown in Figure 2a. In addition, truncated-triangle, hexagon, and butterfly-shaped flakes are formed on the covering SiO_2_/Si substrate as well (as shown in Appendix A), which was observed in previous CVD-grown WS_2_ and MoS_2_ and explained from the point of the changes in the M:S (M = Mo, W) ratio of the precursors within the growth interface [6,31,42,43].

In order to accurately determine the thickness of the as-grown WS_2_ sample, AFM was performed in tapping mode. Figure 2c,d show the AFM image and its corresponding height profiles of an equilateral triangular WS_2_ flake with uniform contrast under OM. It can be clearly observed that there are numerous small particles with several hundreds of nanometers in length and dozens of nanometers in height along the triangle edges. This phenomenon was mentioned by a previous report [37], and it is probably caused by the continuous attachment of the precursors to the growing edge. The thickness of the as-grown WS_2_ flake determined by the height profile is ~0.8 nm, which is consistent with the reported thickness of monolayer WS_2_ [36,44], corroborating the monolayer nature of the triangular WS_2_ flake with uniform contrast obtained under OM. To further identify the atomic structure of our sample, the triangular WS_2_ was transferred to a holy carbon-coated copper grid via the wet transfer method and analyzed by TEM. Figure 2e shows the high resolution TEM image of our WS_2_ sample, which clearly shows the hexagonal ring lattice consisting of alternating tungsten and sulfur atoms. Furthermore, the corresponding selective area electron diffraction (SAED) displayed in Figure 2f reveals only one set of diffraction spots, demonstrating the single-crystal nature of our WS_2_ sample. The inter-planar distances of (100) and (110) planes deduced from high resolution TEM measurement are ~0.269 nm and 0.155 nm, respectively, which coincide with the previous reported results of CVD-grown monolayer WS_2_ [40].

As a powerful and nondestructive tool, the Raman spectrometer has been widely employed to study the properties of TMDs, such as determination of the layer number [45,46,47], electrostatic doping [48], assessment of crystallinity [49,50], as well as internal and external strain [51,52]. Figure 3a shows a typical Raman spectrum of an equilateral triangular WS_2_ flake collected in the backscattering geometry at room temperature. As shown in Figure 3a, except the peak at 520 cm^−1^ from Si substrate, the other peaks are attributed to the WS_2_ flake. The strongest peak at ~350 cm^−1^ can be fitted well with three sub-peaks with a maximum frequency located at 344.7, 350.0, and 354.6 cm^−1^, and they can be identified as the first-order optical mode of ${\;E}_{2g}^{1}\left(M\right)$, the second-order longitudinal acoustic mode of 2LA (M), and the first-order optical mode of $E_{2g}^{1}\left(Г\right)$ (in-plane vibration between sulfur and tungsten atoms as shown in the inset of Figure 3a), respectively, according to the theoretical and experimental studies [45,53]. In addition, strong combination modes of $2\mathrm{LA}\;\left(M\right)-2E_{2g}^{2}\left(Г\right)$ and $2\mathrm{LA}\;\left(M\right)-E_{2g}^{2}\left(Г\right)$ were observed at 294.8 and 321.7 cm^−1^, respectively. The appearance of these strong combination modes is mainly attributed to the strong resonance between the phonon and B-exciton in WS_2_ excited by the 532 nm laser. The peak located at 416.8 cm^−1^ is assigned to the $A_{1g}\left(Г\right)$ mode, which is caused by the out-of-plane vibration between sulfur atoms and is sensitive to the layer number of WS_2_ [46]. Moreover, the intensity of 2LA (M) is much stronger than that of the $A_{1g}\left(Г\right)$ mode, giving rise to an intensity ratio ($R\;={\;I}_{2\mathrm{LA}\;\left(M\right)}/I_{{\;A}_{1g}\left(Г\right)}$) of ~5.6, and the difference between the frequencies of the $A_{1g}\left(Г\right)$ and $E_{2g}^{1}\left(Г\right)$ modes (Δ = $A_{1g}\left(Г\right)-E_{2g}^{1}\left(Г\right)$) is ~62.2 cm^−1^. These characteristics are quite consistent with the reported results for monolayer WS_2_ excited by the 532 nm laser [33,36,41]. To probe the light emission and further determine the layer number of the triangular WS_2_ flake, the room temperature PL spectrum was collected with a 532 nm laser excitation. As shown in Figure 3b, a single and strong peak with a maxima wavelength of 632 nm (~1.96 eV) is observed, which is consistent with the reported PL peak position for monolayer WS_2_ [41,54,55], again confirming the monolayer nature of the as-grown WS_2_.

To investigate the light emission uniformity of the as-grown triangular WS_2_ flake, FL image was captured under excitation of a green light source. Figure 4a clearly shows that the outer region of the triangular WS_2_ flake with an edge length of ~270 μm exhibits intense FL emission, and it becomes weaker towards the inner region. In contrast, the two coalescent small triangles show relatively uniform FL emission across the entire region. This inhomogeneous FL emission in CVD-grown monolayer WS_2_ was observed by other groups. Peimyoo et al. [18] reported a similar observation of the suppression of FL emission for the center region as compared with the edge and they speculated that the suppression was relative to the structural imperfection and n-doping induced by charged defects. Liu et al. [56] observed inhomogeneous FL patterns consisting of alternating dark and bright concentric triangles in monolayer WS_2_, and they attributed the darker region to the high concentration of sulfur vacancies. Recently, Feng et al. [57] reported a novel FL aging behavior in monolayer WS_2_ with a large size, and they attributed that behavior to the partial release of intrinsic tensile strain after CVD growth.

To better understand the heterogeneity of FL emission within the large-area triangular WS_2_, Raman mapping with a step of 3 μm was performed and more than 10,000 data points across the entire flake were collected. As shown in Figure 4b, the distribution of the frequency difference between the $A_{1g}\left(Г\right)$ and $E_{2g}^{1}\left(Г\right)$ modes is in the range of 61 to 62.5 cm^−1^, which is accordance with the reported results for monolayer WS_2_ [33,36,45], demonstrating that the entire triangular WS_2_ possesses a monolayer nature. It has been reported that the $A_{1g}\left(Г\right)$ mode is not only sensitive to the layer number, but also to the electrostatic doping in the typical 2*H*-type TMDs, and it will red-shift and broaden with electron doping [48]. As shown in Figure 4c,d, the frequency and full width of half maximum (FWHM) mappings of the $A_{1g}\left(Г\right)$ mode show relatively uniform features, confirming good uniformity of the crystallinity and electron doping for this large WS_2_ triangle. As a result, the heterogeneous FL emission observed in this WS_2_ triangle is not related to the n-doping. In addition, it has been reported that the frequency of the $E_{2g}^{1}\left(Г\right)$ mode will blue-shift and the intensity of the $A_{1g}\left(Г\right)$ mode will decrease after release of the tensile strain in monolayer WS_2_ [51,57]. To investigate the stress distribution, the normalized intensity mapping of the $A_{1g}\left(Г\right)$ mode and frequency mapping of the $E_{2g}^{1}\left(Г\right)$ mode for this WS_2_ triangle were analyzed, as shown in Figure 4e,f, respectively. It can be clearly seen that the intensity of the $A_{1g}\left(Г\right)$ mode decreases obviously and the frequency of the $E_{2g}^{1}\left(Г\right)$ mode blue-shifts ~1 cm^−1^ for the inner region as compared with the outer region. This indicates that the tensile strain in the outer region is stronger than that of the inner region. By carefully comparing the FL image with the Raman mappings of the $A_{1g}\left(Г\right)$ mode intensity and $E_{2g}^{1}\left(Г\right)$ mode frequency, it can be seen that they match each other well. Therefore, the heterogeneity of FL emission observed in the large WS_2_ triangle is mainly ascribed to the inhomogeneous tensile strain in the WS_2_.

Figure 5 illustrates the FL images of the as-grown WS_2_ flakes with different edge lengths. The FL image of the WS_2_ triangle with an edge length of ~171 μm displayed in Figure 5a possesses similar heterogeneity of the FL emission as the larger WS_2_ triangle observed in Figure 4a. Furthermore, it can be seen that the red large triangle is divided into three smaller and equilateral triangles by three suppressed lines, which is similar to the previous observation in a WS_2_ triangle with an edge size ~100 μm and disappears with aging time [57]. The optical image and Raman mappings of this WS_2_ triangle are shown in Appendix A. It can be seen that the Raman mapping results for this WS_2_ triangle present similar features to the larger one observed in Figure 4a, so that the heterogeneity of the FL emission observed in this WS_2_ triangle is ascribed to the inhomogeneous tensile strain as well. When the edge length of the WS_2_ triangle is reduced to ~95 μm, three lines with slightly weak FL emission can also be observed in the red triangle displayed in Figure 5b, whereas the FL emission shows better homogeneity across the entire triangle. Interestingly, the FL emission becomes more homogeneous and intense when the edge length of the WS_2_ triangle is less than ~32 μm, as shown in Figure 5c. Therefore, it is evident that the monolayer triangular WS_2_ flakes synthesized in this study present distinctly size-dependent FL emission.

Intrinsic tensile strain could be introduced in monolayer MS_2_ grown on SiO_2_/Si substrate during the fast cooling process from the high growth temperature to room temperature due to the higher thermal expansion coefficient of MS_2_ compared with that of silica substrate [57,58]. Feng et al. [57] reported that the intrinsic tensile strain could be partially released from the edge towards the center with the aging time, and consequently concentric FL patterns could be formed in large size (~100 μm) monolayer WS_2_ crystals after 72 h of aging. In this study, all the FL images were captured a week after synthesis, however, we did not detect the concentric FL patterns, probably because the release of intrinsic tensile strain was related to the geometry of the WS_2_ flake and the interaction between WS_2_ and the substrates.

To evaluate the validity and reproducibility of this method, we performed an extra four batches of experiments under the same conditions (950 °C for 5 min). Figure 6 shows the representative optical images of the as-grown WS_2_ flakes obtained from different bathes of experiments under the same conditions. We clearly noted that triangular WS_2_ flakes with an edge length of 250 to 370 μm were readily formed for each batch of the experiment, indicating that the distribution of the solid-sate precursor was controlled well by the drop-casting method. Recently, centimeter scale continuous WS_2_ films with GBs and triangular monolayer WS_2_ flakes hundreds of micrometers (~300 μm) in dimension were grown by direct sulfurization of powdered WO_3_ on SiO_2_/Si substrates, however, the growth time for previous studies was more than 10 min or even more [28,36,57]. In this study, triangular monolayer WS_2_ flakes with an edge length of 250 to 370 μm were effectively formed only within 5 min CVD growth. Therefore, the transportation of precursors could be limited and thus more precursors were able to participate in the synthesis of WS_2_ when a small one-end sealed quartz tube was employed for holding the precursors and substrates simultaneously.

## 4. Conclusions

We demonstrated a facile and controllable method for the synthesis of large-area monolayer WS_2_ flakes by direct sulfurization of powdered WO_3_ on SiO_2_/Si substrates in a horizontal single-zone CVD system. A series of monolayer WS_2_ flakes with an edge length of 250 to 370 μm on SiO_2_/Si substrates were achieved by five independent growth experiments. The morphology, thickness, atomic structure, and light emission of the as-grown WS_2_ samples were characterized using various tools. It was found that the as-grown monolayer WS_2_ flakes exhibit homogeneous crystallinity and size-dependent FL emission. For the WS_2_, triangles with edge lengths less than 95 μm exhibited uniform FL emission while that with larger edge lengths showed heterogeneous FL emission. This heterogeneity of FL emission in large monolayer WS_2_ flakes can be attributed to the heterogeneous release of intrinsic tensile strain after growth. We believe our method could offer an opportunity that opens a new window for the growth of other TMDCs with large areas.

## Figures and Tables

**Figure 1 nanomaterials-09-00578-f001:**
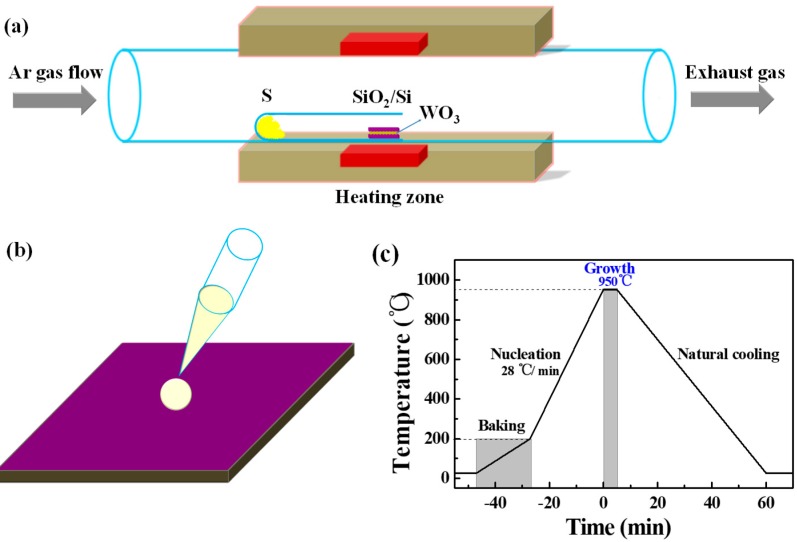
(**a**) Schematic diagram of the horizontal single-zone furnace employed for the synthesis of WS_2_ on SiO_2_/Si substrates. (**b**) Schematic illustration for the drop-casting of WO_3_-ethanol suspension solution onto the SiO_2_/Si substrate. (**c**) Temperature profile adopted for the synthesis of WS_2_ flakes at 950 °C for 5 min.

**Figure 2 nanomaterials-09-00578-f002:**
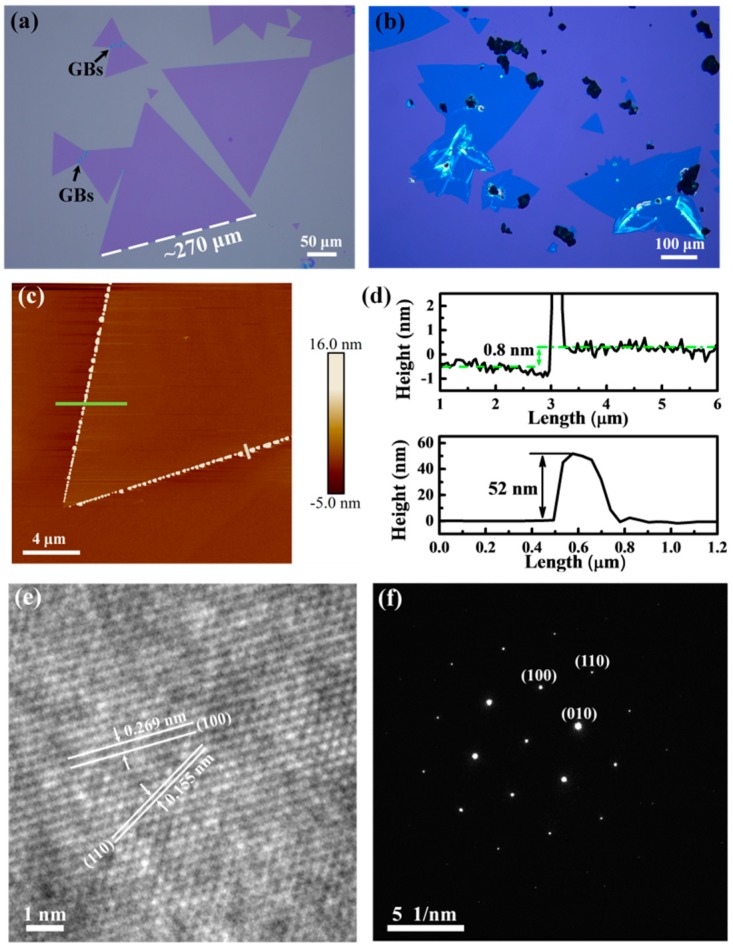
Representative optical images of the as-grown WS_2_ flakes on the covering (**a**) and bottom (**b**) SiO_2_/Si substrates, respectively. AFM image (**c**) and its corresponding height profiles (**d**) of an equilateral triangular WS_2_ flake on SiO_2_/Si substrate. High-resolution TEM image (**e**) and its corresponding SAED pattern (**f**) of the as-grown triangular WS_2_ transferred on a holy carbon-coated copper TEM grid.

**Figure 3 nanomaterials-09-00578-f003:**
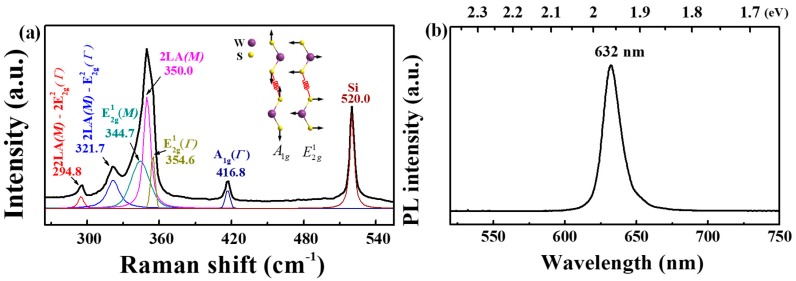
Room temperature Raman (**a**) and PL (**b**) spectra collected from an equilateral triangular WS_2_ flake on SiO_2_/Si substrate with 532 nm excitation. The inset plotted in Figure 3a shows the schematic diagram of the atomic vibrations of $A_{1g}\left(Г\right)$ and $E_{2g}^{1}\left(Г\right)$ modes in WS_2_. It is noted that the black curve is raw data while the color ones are obtained by Lorentz fitting as shown in Figure 3a.

**Figure 4 nanomaterials-09-00578-f004:**
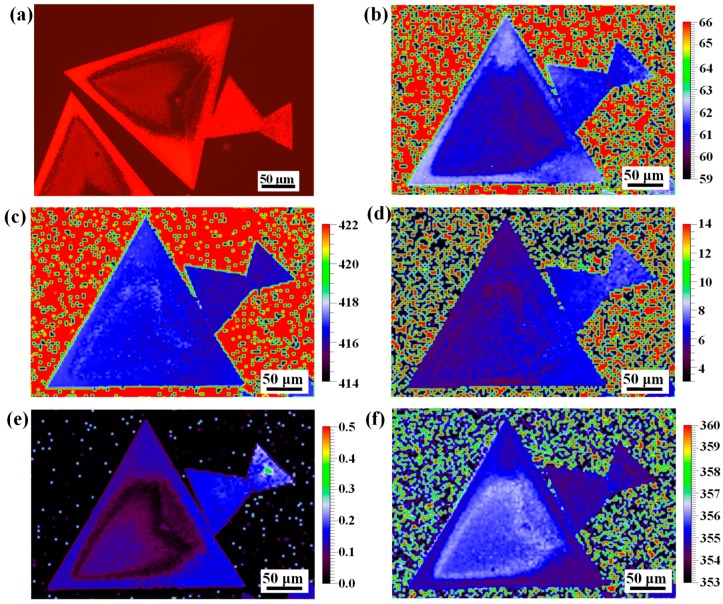
(**a**) FL image of the as-grown WS_2_ flake on SiO_2_/Si substrate, its optical image is shown in Figure 2a. (**b**–**f**) Raman mappings of the specific WS_2_ sample: frequency difference between $A_{1g}\left(Г\right)$ and $E_{2g}^{1}\left(Г\right)$ modes (**b**), frequency of $A_{1g}\left(Г\right)$ mode (**c**), full width of half maximum (FWHM) of $A_{1g}\left(Г\right)$ mode (**d**), normalized intensity of $A_{1g}\left(Г\right)$ mode (**e**), and frequency of $E_{2g}^{1}\left(Г\right)$ mode (**f**).

**Figure 5 nanomaterials-09-00578-f005:**
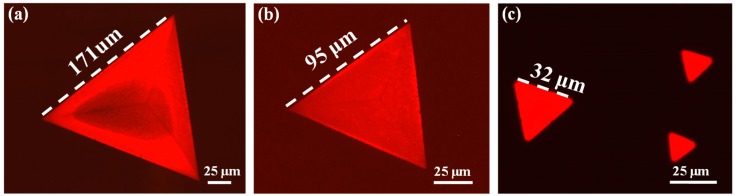
FL images of the as-grown WS_2_ flakes with different edge lengths: ~171 μm (**a**), ~95 μm (**b**), and less than 32 μm (**c**).

**Figure 6 nanomaterials-09-00578-f006:**
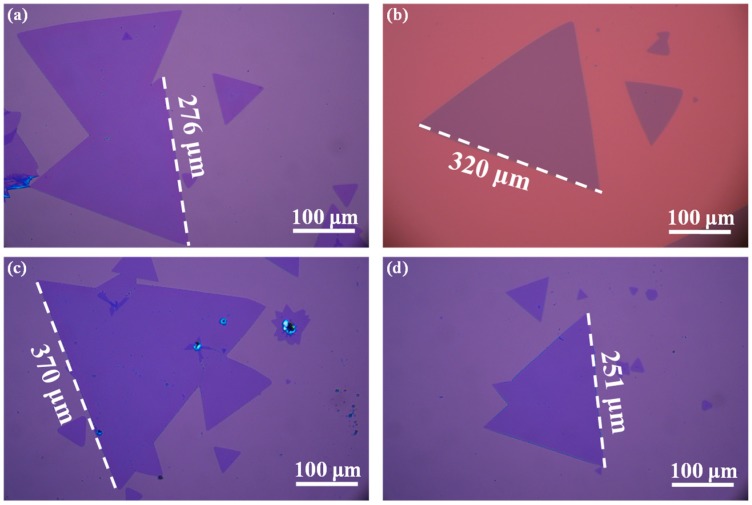
(**a–d**) Representative optical images of the as-grown WS_2_ flakes on the covering SiO_2_/Si substrates obtained from different batches of experiments under the same conditions (950 °C for 5 min.).

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
