# Peer review of "Facile and Controllable Synthesis of Large-Area Monolayer WS_2_ Flakes Based on WO_3_ Precursor Drop-Casted Substrates by Chemical Vapor Deposition"

_nanomaterials, 2019, doi:10.3390/nano9040578_

Reviewer 1 Report

Title:               Facile and controllable synthesis of large-area monolayer WS2 flakes based on WO3 precursor drop-casted substrates by chemical vapor deposition

Authors:         Biao Shi, Daming Zhou, Shaoxi Fang, Khouloud Djebbi, Shuanglong Feng, Hongquan Zhao, Chaker Tlili, Deqiang Wang

Manuscript:   nanomaterials-462682

This is a well written paper describing the growth of WS2 monolayers. The size of the monolayer regions are slightly larger than most reports, although papers are sited that have similar size monolayer flakes.  The main advantage discussed by the authors is the shorter time required to grow the flakes (5 minutes vs 10 or more for competing approaches) and the reproducibility of the technique. The Raman and fluorescence imaging data is very complete and produced results that were consistent with prior reports.

The only question I have is whether there is enough new information for a publication in Nanomaterials.

Author Response

Point 1: The only question I have is whether there is enough new information for a publication in “Nanomaterials”.

Response 1: We are pleased that the reviewer found our work to be interesting. As for your concern, we would like to explain as follows. Firstly, this manuscript focuses on the fabrication and characterization of monolayer WS2, which is quite relevant to the recent special issue "2D Materials and Van der Waals Heterostructures: Physics and Application” for “Nanomaterials”. Secondly, we have conducted a survey of papers published in “Nanomaterials”, until now there are few papers related to 2D materials of WS2. Thirdly, compared to the previous reports, the method developed by us possesses the advantages of saving time, saving precursors and highly reproducible, which opens up a new window for large-area, controllable growth of monolayer WS2. Therefore, we think our work will provide enough new and helpful information for a publication in “Nanomaterials”. 

Reviewer 2 Report

The current manuscript titled "Facile and controllable synthesis of large-area monolayer WS2 flakes based on WO3 precursor drop-casted substrates by chemical vapor deposition" by Biao Shi et al., on the growth of large area WS2 flakes by CVD method is well-written and the results are carefully explained by the authors.

However, I would like the authors to address the following in the manuscript to improve the quality of the manuscript to meet the journal standards.

1. I suggest the authors to discuss on the importance of the current research in terms of the idea, growth technique and the obtained properties of the WS2 flakes in comparison to the published work by Pengyu Liu et al., in Reference 40. Apart from drop-casting WO3-ethanol on the substrate during the growth, how is the current technique different from Ref. 40?

2. How are the properties of the obtained WS2 flakes different from that in Ref. 40 except for the larger size of the obtained nanosheets?

3. I suggest the authors to confirm the mono-layer thickness of the flakes by TEM imaging. I also recommend TEM diffraction for confirming the crystallinity of the flakes. What is the composition of the flakes?

Author Response

Point 1: I suggest the authors to discuss on the importance of the current research in terms of the idea, growth technique and the obtained properties of the WS2 flakes in comparison to the published work by Pengyu Liu et al., in Reference 40. Apart from drop-casting WO3-ethanol on the substrate during the growth, how is the current technique different from Ref. 40?

Response 1: We really appreciate your constructive comments that greatly help us to improve the quality of our manuscript. As we know, Ref.40 cited in this manuscript was the first paper that investigated the impacts of different growing parameters on the morphology of the WS2 films, from which we have deepened our understanding of the WS2 growth. At the beginning, we almost utilized the same strategy as Ref.40 for WS2 synthesis. However, we find that the distribution of the solid-phase precursors in ceramic boats is not easy to control, which could significantly influence the concentration of gaseous species on the growth interface. Furthermore, due to the difference in melting points of sulfur and WO3, it is difficult to simultaneously and precisely control the evaporation of these two precursors and the subsequent transportation to the growth interface. As a result, the dimension and morphology of the as-grown WS2 always present significant difference even under the same growth parameters. To overcome these drawbacks, we disperse the powdered WO3 in ethanol to form suspension solution, then we use pipette to drop-cast the solution onto the SiO2/Si substrates, by which the distribution of WO3 is effectively controlled batch by batch. Furthermore, we use a small quartz tube sealed at one-end to load precursors and substrates simultaneously, which is quite different from Ref.40. This small one-end sealed quartz tube can limit the transportation of gaseous species and then more precursors can participate in the synthesis of WS2. By this method, we can obtain monolayer WS2 with edge length up to 250 ~ 370 μm within 5 min CVD-growth. The growth time for similar size monolayer WS2 is much shorter in our method than that in Ref.40.

Point 2: How are the properties of the obtained WS2 flakes different from that in Ref. 40 except for the larger size of the obtained nanosheets?

Response 2: By carefully comparing the results with Ref.40, the properties of the obtained WS2 flakes in this study are mainly different as follows. Firstly, the strong PL peak position of monolayer WS2 is slightly different. The PL peak position is observed at 627 nm in Ref.40while it is observed at 632 nm in this manuscript. This difference in PL peak could be caused by the difference in stress, defects and impurities within the monolayer WS2 flakes, and it has been well investigated in Ref.4154 and 55. Secondly, the Raman mapping images present uniform features in Ref.40, but the authors did not study the size effect on the Raman mapping. For our results, the uniformity of the Raman mapping results strongly depends on the edge length of the as-grown WS2 flakes, and we attributed that heterogeneity to the inhomogeneous tensile strain within the WS2. Thirdly, the fluorescence emission for the monolayer WS2 with edge length of ~ 30 μm is suppressed in the inner region in Ref.40. However, the fluorescence emission is quite uniform for our monolayer WS2 with similar edge length. Furthermore, we observed apparently size-dependent fluorescence emission in our monolayer WS2, and we attributed it to the heterogeneous release of intrinsic tensile strain after growth.

Point 3: I suggest the authors to confirm the mono-layer thickness of the flakes by TEM imaging. I also recommend TEM diffraction for confirming the crystallinity of the flakes. What is the composition of the flakes?

Response 3: The thickness of the as-grown WS2 in this manuscript is ~ 0.8 nm measured by AFM, which is consistent with the result of ~ 0.82 nm in Ref.40. Furthermore, the monolayer nature of the as-grown WS2 in this study has been verified by PL and Raman, which are widely used to identify the layer number of layered materials, such as graphene and MoS2. Furthermore, we would very much like to consider the characterization of TEM, which can provide crystal structure, defects, and composition information for the as-grown WS2. However, our institute has not TEM facility yet, we only can send samples to other universities for testing. Until now, we have prepared several WS2 samples for TEM characterization, but most of them have been damaged during the delivery process and we could not get perfect TEM results (Fig.1). We are trying to figure out this issue in our future work.

Fig.1 High-resolution TEM image (a) and its corresponding SAED pattern (b) of the freely suspended monolayer WS2 on a TEM grid

Reviewer 3 Report

Manuscript # Nanomaterials-462682: “Facile and controllable synthesis of large-area monolayer WS2 flakes based on WO3 precursor drop-casted substrates by chemical vapor deposition”

B. Shi and co-authors have demonstrated that WS2 monolayers can be grown up to the lengths of several hundred nanometers controllably by their one-step CVD method incorporating the drop-casting process. The high quality of CVD-grown WS2 monolayer have been confirmed thoroughly by various experimental techniques including optical microscopy, Raman and PL spectroscopies, and FL imaging. Considering that the material synthesis and characterization have been performed rigorously and the invented technique is useful for the community, some minor concerns should be addressed for the publication in Nanomaterials.

My detailed concerns are addressed as follows:

1.      There are some grammatical errors throughout the manuscript. It should be proof-read thoroughly.

2.      In the Fig. 1 (c) and Fig. 3 (a) & (b), it would be hard for readers to read fine letters.

3.      The Fig. 2 (c) is so crowded that the line profiling graphs cannot be read easily.

4.      In the Fig. 4, units are missing for all the scale bars.

5.      In the Fig. 5, there are no scale bars and units for the FL images.

6.      Is there any particular reason why the results obtained from random flakes are presented for the multiple experiments in the main text? It would be better for authors’ discussion and analyses focus on the results measured on a couple of identical representative flakes throughout the manuscript.

Author Response

Point 1: There are some grammatical errors throughout the manuscript. It should be proof-read thoroughly.

Response 1: Thank you for your comment. The grammatical errors have been corrected and highlighted by red words in the revised manuscript. Some of them list below:

Line 43 “atomically thin WS2 still a big” has been corrected to “atomically thin WS2 is still a big”;

Line 50 “electrical property” has been corrected to “electrical properties”;

Line 114 “substrate and sequentially washed…”has been corrected to “substrate and was sequentially washed…”;

Line 128  “ensuring the substrates are exactly located” has been corrected to “ensuring the substrates were exactly located”;

Line 153 “caused by coalescing between adjacent triangles” has been corrected to “caused by coalescence of adjacent triangles”;

Line 196  “The peak at 416.8 cm-1” has been corrected to “The peak located at 416.8 cm-1”;

Line 237  “will decrease after the release of the tensile strain…” has been corrected to “will decrease after release of the tensile strain”;

Line 272  “intrinsic tensile strain is related to” has been corrected to “intrinsic tensile strain was related to”.

Point 2: In the Fig. 1 (c) and Fig. 3 (a) & (b), it would be hard for readers to read fine letters.

Response 2: We agree with the reviewer and the Fig.1 (c) and Fig. 3 (a) & (b) have been modified in the revised manuscript, especially for the captions font size. The modified figures are listed as follows

Fig.1 (c) Temperature profile adopted for the synthesis of WS2 flakes at 950 for 5 min.

Fig. 3 Room temperature Raman (a) and PL (b) spectra collected from an equilateral triangular WS2 flake on SiO2/Si substrate with 532 nm excitation.

Point 3: The Fig. 2 (c) is so crowded that the line profiling graphs cannot be read.

Response 3: The insets in Fig.2(c) have been plotted independently in the revised manuscript.

Fig.2. AFM height image (c) and profiles (d) of an equilateral triangular WS2 flake on SiO2/Si substrate.

Point 4: In the Fig. 4, units are missing for all the scale bars.

Response 4: The scale bars have been added on the Raman mapping images in the revised manuscript.

Fig.4 FL image and Raman mappings of as-grown WS2

Point 5: In the Fig. 5, there are no scale bars and units for the FL images.

Response 5: We have added the scale bars and units for the FL images in the revised manuscript.

Fig.5 FL images of the as-grown WS2 flakes with different edge length

Point 6: Is there any particular reason why the results obtained from random flakes are presented for the multiple experiments in the main text? It would be better for authors’ discussion and analyses focus on the results measured on a couple of identical representative flakes throughout the manuscript.

Response 6: The purpose of the multiple experiments is to show the reproducibility of this developed method. So the representative optical images of the as-grown WS2 flakes from different batches of experiments are collected in Fig.6.

Round  2

Reviewer 2 Report

Thank you for the authors´ response.

I suggest that the authors to revise the manuscript to reflect the significance and impact of the present synthesis method and the obtained material properties. Justify the novelty in the synthesis method and characterization to meet the journal standards. TEM characterization is to be included and described in detail in the manuscript.

Author Response

We really appreciate you carefully reviewed the manuscript again. According to your suggestions, we have revised the manuscript and added the TEM characterization results in the revised manuscript. Compared to the previous reports, the method developed by our research group has the advantages as follows. Firstly, the present method can save the WO3 precursor. The amount of WO3 used in this work is ~15 µg (10 μl × 1.5 mg/ml), while the amount of W-precursor used in previous reports is in the range of milligram. Secondly, the present method can also save the growth time. Triangular monolayer WS2 flakes with edge length of 250 ~ 370 μm are readily synthesized within 5 minutes of growth in this study, which is shorter than that of the previous reports (with similar size of monolayer WS2). Thirdly, the present method is highly reproducible, and it is confirmed by 5 independent experiments under the same conditions.

Reviewer 3 Report

In my opinion, all of my concerns have been well addressed. Further, the revised manuscript appears well-organized and authors’ arguments become stronger and clearer. I would like to recommend publication of this manuscript.  

Author Response

Thanks very much.

Round  3

Reviewer 2 Report

Thank you for considering the revision and implementing the suggested changes.